# Conditional Generative Models for Dynamic Trajectory Generation and Urban Driving

**DOI:** 10.3390/s23156764

**Published:** 2023-07-28

**Authors:** David Paz, Hengyuan Zhang, Hao Xiang, Andrew Liang, Henrik I. Christensen

**Affiliations:** Autonomous Vehicle Laboratory, Contextual Robotics Institute, University of California San Diego, La Jolla, CA 92093, USA; hyzhang@eng.ucsd.edu (H.Z.); haxiang@g.ucla.edu (H.X.); aliang@ucsd.edu (A.L.)

**Keywords:** autonomous driving, perception, scene understanding, generative models, global planning, coarse maps, HD maps, semantic maps

## Abstract

This work explores methodologies for dynamic trajectory generation for urban driving environments by utilizing coarse global plan representations. In contrast to state-of-the-art architectures for autonomous driving that often leverage lane-level high-definition (HD) maps, we focus on minimizing required map priors that are needed to navigate in dynamic environments that may change over time. To incorporate high-level instructions (i.e., turn right vs. turn left at intersections), we compare various representations provided by lightweight and open-source OpenStreetMaps (OSM) and formulate a conditional generative model strategy to explicitly capture the multimodal characteristics of urban driving. To evaluate the performance of the models introduced, a data collection phase is performed using multiple full-scale vehicles with ground truth labels. Our results show potential use cases in dynamic urban driving scenarios with real-time constraints. The dataset is released publicly as part of this work in combination with code and benchmarks.

## 1. Introduction

Autonomous vehicles architectures today depend heavily on the high-definition (HD) maps, especially on the planning aspect. Figure 1 shows an example architecture where a lane-level HD map is used in both global planning and motion planning. For global planning, the shortest path can be estimated by running A* [1] on the lane graph. The path, which consists of centimeter-level accurate waypoints and speed limits, along with the objects derived from perception and traffic control information from HD maps, is then utilized in the motion planning stage. The motion planning stage determines the vehicle’s driving state (e.g., forward, following, and stop) and constrains the vehicle speed.

However, the architectures that rely on HD maps are not scalable as HD maps are costly to create and require constant maintenance. These maps with accurate lane-level geometry, speed limit, and traffic control information often require human labeling or inspection, and infer additional overhead when expanding the maps to new areas. Additionally, road closures and construction sites can render maps obsolete and thus previously mapped areas need to be maintained constantly.

Existing research attempts to find solutions to this problem from two directions: offline automatic HD map generation and online HD map generation. Nevertheless, the works that focus on automatic HD map generation using aerial imagery [2,3] or ground vehicle data [4] are offline or are limited by coverage and still require constant updates. On the other hand, building HD maps online [5,6,7,8] reduces the priors needed. Recent works explore online vectorized HD maps [6,7,8] but the number of map elements generated is still limited for them to be integrated with traditional HD map based architectures. Moreover, performance of these methods on real-time autonomous driving frameworks currently remains an open research question.

To tackle these problems, we propose an alternative strategy and architecture to leverage coarse maps and local semantic representations for planning, as shown in Figure 1. Coarse maps, such as open-sourced OpenStreetMaps (OSM) and proprietary maps like Google Maps, are lightweight. These maps generally provide road segments and intersection labels but lack lane and trajectory information with centimeter-level geometry. They offer scalable solutions for global planning. The generated global plan encoded in the graph captures information such as “turn right at the next intersection”. This high-level global plan requires context to guide the vehicle driving. We use a local semantic map to provide such context. With the global plan and local context, we focus on generating a local trajectory dynamically that can be referenced by the downstream motion planning task.

This work explores methodologies for the dynamic trajectory estimation task by utilizing coarse global plan representations. To include high-level instructions in the planning process, such as turning left or turning right at intersections, we evaluate various rasterized and vectorized representations that are generated by using the lightweight and open-source OSM. We then utilize these representations and propose a conditional generative model that can explicitly capture the multimodal features of urban driving. A benefit of our approach is that it can be utilized with existing autonomy stacks. In Figure 1, we can observe the key differences between an HD-map-based planner and a coarse map based planner. Although the input representations and priors vary, the reference paths generated from the planners can serve downstream application tasks, such as motion planning. To assess the effectiveness of the proposed models, we conduct multiple data collection phases that involve multiple full-scale vehicles with ground truth labels.

This paper unifies two of our previous contributions [9,10] by providing additional details on experiments, strategy, and data collection. We additionally make our complete datasets, code repositories, and benchmarks open and publicly available.

In Section 2, we discuss related work in the field and the relevance of recent real-time strategies for urban driving navigation. The strategies for dynamic trajectory generation and planning are then introduced in Section 3. We specifically discuss the technical aspects of the method, implementations and code released. We then provide an in-depth description of the data collection process, metrics, and ablation studies in Section 4. Lastly, we discuss results and make note of potential future work directions in Section 4.4 and Section 5, respectively. In summary, our contributions include:A formulation for dynamic trajectory generation utilizing nominal global plan representations and local semantic scene models.We provide an in-depth analysis on the performance benefits of using graphical methods to represent coarse plans.We release two datasets with over 13,000 synchronized global plan and semantic scene representations. In contrast to existing datasets, our data can enable research directions for path planning with fewer priors.The code repositories for the nominal planner and dynamic trajectory generation are made open source and publicly available (https://github.com/AutonomousVehicleLaboratory/gps_navigation, accessed on 26 July 2023,https://github.com/AutonomousVehicleLaboratory/coarse_av_cgm.git, accessed on 26 July 2023).

## 2. Related Work

In this section, we review related research on HD map generation. We then look at research related to two key components of our approach: lightweight maps and scene representations.

### 2.1. HD/Vector Maps

Over the years, many different methodologies have been developed for autonomous navigation in urban environments. Many of these methodologies use HD maps to facilitate the process of path planning. Examples of classical motion planning and control architectures that utilize HD maps include Autoware [11] and Apollo [12]. These types of architectures have shown promising results for micro-mobility applications over relatively extended periods of time [13]. Similarly, end-to-end strategies [14,15,16,17,18,19,20,21] have increasingly gained popularity in the research community. These methods generally rely on large training datasets [22] or simulation environments [23,24] for development and validation.

Recent work additionally explores building HD maps automatically to remove the limit of scalability imposed by human labeling. The data representations are often captured from aerial imagery or ground vehicles. Works from aerial imagery [3] can be limited by the availability of images and will not be able to capture traffic signs; however, vehicles can naturally provide more effective coverage. Homayounfar et al. [5] and Zhou et al. [4] build lane-level HD maps in highway and urban areas, respectively. VectorMapNet [6] proposes to predict vectorized representations directly, and MapTR [7] improves prediction performance by using permutation invariant representations and loss. TopoNet [8] further considers the association between lane-to-lane and lane-to-traffic elements (signs and traffic lights), which are essential elements to downstream tasks. However, in contrast to these efforts, our work focuses on the direct trajectory prediction task that is amenable for downstream applications instead of the mapping process itself.

### 2.2. Lightweight Maps

Methods that reduce the dependencies and priors on maps have also been an active area of research in recent years. The underlying representations that encode turn-by-turn directions for the global planning task are comparable to Google’s proprietary maps or the publicly available OSM. For instance, generative models have been developed to encode the multimodal characteristics of driving implicitly [25] and explicitly [26] by utilizing coarse maps, such as OSM [27]. An advantage of these representations is that the overhead associated with curating and updating maps can be reduced by relying more on raw sensor data, such as camera image streams. Other related work involves utilizing a discrete action space, such as “turn left,” “turn right,”, or “go straight”, to shift towards “mapless” strategies. This idea also focuses on minimizing the priors and reliance on maps; examples include Light Detection and Ranging (LiDAR) based methods [28] and camera-based works [29] by using one-hot encoding to represent the desired action specifically for intersections.

### 2.3. Scene Representations

Recent developments in the semantic segmentation and sensor fusion literature have enabled methods for scene understanding that seek to build scene representations with highly detailed and accurate localization of road features without manual input. For example, [30,31] focuses on generating 2D bird’s eye view semantic scene representations from monocular camera streams. While these present advantages for real-time scene understanding, they still present considerable limitations in terms of occlusions and localization errors of road futures. Alternative offline methods [32] can address some of these limitations through a spatiotemporal fusion process that leverage camera and LiDAR fusion. Nevertheless, these efforts still require additional post-processing to extract lane-level trajectories that can be ingested by downstream modules. To address this challenge, our work seeks to align coarse representations (similar to Section 2.2) with 2D bird’s eye view semantic maps to estimate traversable lane-level trajectories. The motivation for utilizing coarse maps is centered around providing high-level global planning information with minimum priors and the use of semantic maps is to utilize scene representations that are accurate, up-to-scale, and that can be generated automatically.

## 3. Methods

In this section, we introduce the approach for aligning nominal global plans with semantic scene representations to predict egocentric trajectories that align with both the global plan and the lane-level features provided by the semantic map. We explore various representations for encoding the global plans, including rasterized and vectorized representations. The semantic scene representations are generated by utilizing a probabilistic semantic process with accurate localization to provide the context in an egocentric frame. Finally, the alignment is performed by utilizing a Conditional Variational Autoencoder (CVAE) to model the distribution of trajectories that can be executed given the global plan and semantic scene representation.

We introduce the global planner with its rasterized and graph representations in Section 3.1. Then in Section 3.2 the local scene representation is described. Finally, we introduce the conditional generative models and their loss functions used to generate the nominal trajectory in Section 3.3.

### 3.1. Global Planning

A global planner is implemented based on traditional graph search algorithms. This is performed by utilizing Global Navigation Satellite System (GNSS) traces to fetch and download open-source OSM data (https://www.openstreetmap.org/, accessed on 26 July 2023). Each OSM is saved in an Extensible Markup Language (XML) format, which encodes map connectivity information. This format is parsed and post-processed to populate a graph with full road connectivity, where distance and road element information is preserved. In practice, we perform a projection from a geographic coordinate system into a Cartesian space by using the Haversine distance formula to approximate the relative distance on a sphere (earth). This approximation allows us to characterize a graph search strategy that is performed on a plane and assign weights based on units of meters, and also perform an egocentric transformation based on the position and orientation of the agent on the map. An example of this transformation can be seen in Figure 2, where we visualize the vehicle-centric rasterized (left) and vectorized (right) representations that are utilized in our work.

The rasterized representation is generated by defining a blank image canvas and drawing straight line segments that represent the road. In this representation, the road segments are represented by white lines and the planned road segments are denoted in green. The size of the image is 200 px × 200 px with a 0.5 m/px resolution. Given that this is a raster, the information provided in Cartesian space is discretized to convert the map information into an image and encoded using a Convolutional Neural Network (CNN)-based encoder.

In contrast, the graphical representation of the road network elements is directly extracted from the original graph by performing a local search within the vehicle location and state. The nearby elements are additionally decorated with various attributes from nearby features on a per-node basis, such as stop signs, crosswalks, and traffic lights. This representation considers every node in the graph as a 3D vector—where the first two dimensions represent the 2D coordinates of the node element in an egocentric frame and the last dimension denotes if the node corresponds to a traffic signal, pedestrian crosswalk, or a stop sign. This compact global plan $g_{m}={\left\{\left(w_{x}^{d},w_{y}^{d},w_{f}^{d}\right)\right\}}_{d=1}^{D}$ is represented by a 2D tensor of shape D×3, where D=40 waypoints are utilized to represent the global plan. The first $\frac{D}{2}$ waypoints represent the trajectory directly behind the vehicle and the last $\frac{D}{2}$ waypoints represent the relative plan ahead of the vehicle. A self-attention mechanism [33] is then applied on this input representation, as shown in Equation (Equation 1); where C=3 and Q, K, V are linear projections of the path. In Section 4.4.2, we perform an ablation study on the most relevant global planner features.
(1)$$\mathrm{SA}\left(g_{m}\right)=g_{m}+\mathrm{softmax}\left(\frac{QK^{T}}{\sqrt{C}}\right)V$$

An important note in the design of our approach for the global planner is that the egocentric transformation is performed after a global plan is determined with respect to the map frame. This permits the egocentric representation to encode relative target information from a local perspective up to a fixed horizon, which applies to both representations: the rasterized and vectorized representations. Our implementation for the planner is open-source and implemented using the Robot Operating System (ROS) framework (https://github.com/AutonomousVehicleLaboratory/gps_navigation, accessed on 26 July 2023).

### 3.2. Semantic Scene Representation

Although the data provided by the OSM-based planner can provide certain information about the underlying road geometry for a particular scene, various features and important semantic cues, such as lane-level connectivity information and boundaries, are not available. To incorporate this information, an additional representation that can provide context is necessary. Hence, we leverage a spatio-temporal camera-LiDAR fusion method to generate local semantic scene representations in an egocentric perspective. This process consists of a three-stage pipeline which is used to project semantics from image based semantic segmentation onto a 2D bird’s-eye view (BEV) as outlined in Figure 3.

To represent the 2D BEV semantic map, we use an occupancy grid-based representation with three dimensions: (H×W×C). The height *H* and width *W* of the grid represent spatial dimensions, and the three channels *C* represent the color of different semantic classes, such as roads, sidewalks, and buildings. When a semantic point cloud is created, it is projected onto the semantic occupancy grid. Each point in the point cloud is then assigned to the corresponding grid cell and the semantic label of each point is then determined by utilizing a probabilistic approach based on the known image semantic segmentation confusion matrix. Additional details can be found in [32].

In practice, these maps are generated automatically offline and utilized during training with the known vehicle poses (estimated from localization). The position and orientation of the agent is used to extract local regions with respect to the rear-axle of the vehicle and perform a rotation such that the semantic features are aligned with the longitudinal axis of the vehicle, i.e., the front of the vehicle points up in the 2D BEV local semantic scene representation and the center of the image represents the rear-axle of the robot.

Finally, we utilize a sequence of CNN layers to process the local semantic scene representation, as shown in Figure 4.

### 3.3. Conditional Generative Models

Motivated by the multimodal characteristics of urban driving navigation, we formulate the dynamic trajectory generation process by utilizing a Conditional Variational Autoencoder. A CVAE can be considered a conditional generative model that is capable of capturing and approximating a conditional distribution explicitly, namely $p\left(y|s,g\right)$. In this work, we let $y=\{\left(x^{1},y^{1}\right),\cdots ,\left(x^{H},y^{H}\right)\}$ represent the lane-level trajectory of interest and s and g are the local semantic scene representation and the local global plan, respectively. To simplify notation and the derivation for a CVAE, we let m={s,g} jointly represent the semantic and local plan information and rewrite the probability distribution as $p\left(y|m\right)$. An important step in the derivation for CVAEs as described in [34] involves approximating the distribution by introducing a latent variable z that is drawn from $p\left(z|m\right)$. If z is assumed to be discrete, $p\left(y|m\right)$ can be rewritten by marginalizing over z, as shown in Equation (Equation 2).
(2)$$p\left(y|m\right)=\sum_{z\in Z}p_{\phi }\left(y|m,z\right)p_{\theta }\left(z|m\right),$$

To characterize the objective function desired as part of the optimization process, the CVAE approach also introduces a recognition model denoted by $q_{\psi }\left(z|m,y\right)$. This distribution is utilized only during the training process of the model and is not used in testing. The intuition behind this distribution is that during training, it has access to ground truth trajectory information (y) and can be jointly optimized with $p_{\theta }\left(z|m\right)$ by utilizing the Kullback–Leibler (KL) divergence. This process guides $p_{\theta }\left(z|m\right)$ in the training process and can be used at test time without ground truth. The pipeline for training and testing is shown in Figure 4; where we base our architecture in a similar way as [35]. The overall objective function used in practice is derived as a combination of the standard CVAE formulation with an added Mean-Squared-Error (MSE) term to minimize displacement error, as shown in Equation (Equation 3).
(3)$$L=-E_{q}\left[\log p_{\phi }\left(y|m,z\right)\right]+\mathrm{KL}\left[q_{\psi }\left(z|m,y\right)∥p_{\theta }\left(z|m\right)\right]+\frac{1}{H}\sum_{h=1}^{H}{∥y^{h}-{\hat{y}}^{h}∥}^{2}$$

At test time, we decode a trajectory by using the mode $z^{*}$ that maximizes $p_{\theta }\left(z|m\right)$. In other words, we utilize Equation (Equation 4) to decide which of the |Z| modes to utilize to make a prediction. The trajectories are regressed by a Gated Recurrent Unit (GRU) [36] module, which parameterizes *H* Bivariate Normal Distributions.
(4)$$z^{*}=\arg\max_{z}p_{\theta }\left(z|m\right)$$

## 4. Experiments and Data

Our approach utilizes synchronized global plan and local semantic scene representations to condition the CVAE. The synchronized data samples are generated as part of a data collection process and can be visualized in Figure A1. First, we set a destination using the OSM planner described in Section 3.1. This planner utilizes GNSS and inertial measurement unit (IMU) data to estimate and correct the position of the ego vehicle over time and perform the necessary egocentric transformations. The output generated from the planner consists of a 2D raster and the graph representation (Figure 2). Similarly, a local semantic scene representation is generated in an egocentric frame by utilizing a LiDAR-based localization method. These representations are extracted from full semantic maps. Both representations are synchronized based on the nearest timestamps. The intrinsic and extrinsic parameters that are used for estimating egocentric transformations are estimated offline utilizing standard calibration methods such as checkerboards and plane fitting methods with feature matching strategies. Additional details on the process for calibration and localization can be found in [13].

We use the reported ego-vehicle poses from a LiDAR-based registration method to characterize ground-truth trajectories. Multi-sensor fusion estimates are generally also accurate enough to provide pose estimates as additionally noted in [37]. However, since the position and the orientation of the vehicle is reported at 10 Hz, the data points recorded during data collection can be inconsistently spaced apart. Hence, we interpolate as noted in Figure A1 and sample new trajectories during training and evaluation. This allows us to sample trajectories with varying waypoint density and analyze the limitations of each.

### 4.1. Datasets

With the data collection process described, various datasets are curated at the UC San Diego campus. The NominalScenes dataset [9] includes urban driving data with global plan and semantic maps in various driving scenarios such as curved roads, loops, and intersections. On the other hand, the IntersectionScenes dataset [10] includes global plans and semantics for intersection-specific scenarios such as three-way and four-way intersections. We use these datasets to train and test the performance of our methods. We additionally make our implementation and benchmark publicly available (https://github.com/AutonomousVehicleLaboratory/coarse_av_cgm.git, accessed on 26 July 2023). The visualizations of the point cloud maps used in the semantic mapping process are shown in Figure 5 alongside the output semantic maps.

### 4.2. Platform and Hardware Requirements

The data collection process is completed using two identical vehicle platforms with similar sensor arrangements. A high-level overview of the sensor arrangement and data collection process is shown in Figure A1. Each of the platforms comprises six Gigabit Ethernet Mako G-319 cameras, a Velodyne VLP-16 LiDAR, a Garmin GNSS system, and an IMU. The computer platform on board includes an Intel Xeon E3-1275 CPU, an NVIDIA GTX 1080Ti GPU, and 32 GB RAM. This system is used to collect the sensor data using the ROS framework’s bag file format.

The LiDAR and the front two cameras are utilized in the semantic mapping framework. This mapping process is performed offline using an Intel i9-7900 CPU, an NVIDIA Titan Xp, and 128 GB RAM. Once the semantic map is generated for the regions of interest, the map can be reused for training or online inference. The GNSS and IMU are used as part of the OSM planner; the plans are automatically synchronized with the local semantic maps based on nearest Unix Epoch timestamps. The training process for the CVAE is additionally performed offline using the same system. The hardware requirements for the training process are relatively low and can be met with less than 12 GB GPU VRAM.

### 4.3. Metrics

We measure the quality of each trajectory produced based on two criteria: Driveable Area Compliance (DAC) and Displacement Error (DE). DAC evaluates the model’s capacity to generate trajectories that stay within drivable regions. This measurement is derived by averaging the trajectories that coincide with drivable areas, including crosswalks, lane markings, and road surfaces as defined in the local semantic map. If any waypoint of a trajectory overlaps with a sidewalk or vegetation area, it is deemed non-compliant. We present the error associated with half of the trajectory (DAC${}_{HALF}$), as well as the entire predicted trajectory (DAC${}_{FULL}$), as shown in Equations (Equation 5)–(Equation 11). Both metrics characterize compliance in a range between [0, 1] by utilizing an indicator function to evaluate if a waypoint prediction ${\hat{y}}_{i}^{h}$ lies within the semantic set of drivable regions $C_{d}$.

We additionally employ metrics commonly used in the field of road user prediction research to characterize the performance of trajectories: Average Displacement Error (ADE) and Final Displacement Error (FDE) [38,39]. These metrics allow us to assess the average error of each trajectory across all *H* waypoints (ADE) and the average error specifically related to the last predicted waypoint (FDE). To provide a more comprehensive analysis, we extend the evaluation of ADE by measuring the error associated with half of the trajectory (ADE${}_{HALF}$), considering that the waypoints closest to the autonomous agent are executed first during navigation. The worst-case errors are also captured by calculating the average maximum displacement error (MDE) along each predicted trajectory.

### 4.4. Results

For all the experiments performed, we set the number of discrete states within the CVAE to 12, i.e., |Z|=12. The rationale for choosing 12 as the number of distribution modes is not only motivated by performance results but also by the various navigation behaviors including (i) making left turns, (ii) making right turns, (iii) lane following, (iv) driving straight across intersections, (v) driving along curved roads, and (vi) making u-turns. This equates to utilizing two modes to model each driving behavior explicitly.

#### 4.4.1. Ablation Study: Waypoint Density

In our initial experiments, we compare trajectory predictions with varying waypoint densities up to a 30 m horizon. The first type H10 is defined by 10 waypoints spaced 3 m apart and the second H15 defines 15 waypoints spaced 2 m apart. The performance of these two characterizations of the model are shown in Table 1. From the results, we observe that increasing the number of waypoints regressed also increases the likelihood of encountering compound errors. We hypothesize that this is due to the dependency of the next waypoint prediction on the previous cell state within the GRU decoder of the network. Therefore, for the remainder of the experiments, we utilize H10 as the underlying trajectory representation.

#### 4.4.2. Ablation Study: Rasterized and Graph Representations

To further evaluate the performance implications of the underlying global plan representations, we compare the rasterized and the graph-based global plan representations discussed in Section 3.1. In Table 2, we evaluate four different models using the NominalScenes dataset and benchmark. The first method evaluated utilizes the rasterized global plan, and the last three leverage the graphs directly while aggregating various node features. For graphical models, we analyze the value of incorporating information about the road segments traversed (*P*), the planned road segments to be traversed (*F*), traffic signals and stop signs (*S*), and crosswalks (*C*). In these experiments, we observe that the attention mechanism that makes use of all node features except crosswalks outperforms the rasterized method and the methods that only make partial use of the node features. However, we note that not all the node features necessarily boost performance. For example, crosswalk features slightly deteriorate performance when we compare the two models that make use of the full history, planned trajectory, and traffic signals and stop sign features. While this may be unexpected, we find that OSM crosswalk information is not unique to intersections and as a result may present difficulties distinguishing between intersections with crosswalks and straight road segments with crosswalk features. On the other hand, this is not the case when we incorporate stop signs and traffic signals, since they are consistently defined at intersections.

In a subsequent set of experiments, we evaluate the performance of our methods for intersection-specific scenarios using the IntersectionScenes dataset; these scenarios include left turns, right turns, and driving straight across intersections for three-way and four-way intersections. The results consistently show that explicitly utilizing the graphical representation can boost performance.

On average, we observe an 1.5 m error across the full trajectory generated, and an average error of 3.1 m associated with the last waypoint of every trajectory predicted. We additionally find that the worse case predictions average to 3.2 m. Similarly, DAC indicates that 90% of the trajectories estimated overlap with a drivable region, including pavement, crosswalks, and lane marks. For additional details, please see Table 3.

In contrast, the errors associated with the first half of each trajectory predicted are considerably lower. These results are particularly more relevant given that a motion planner will utilize the nearby waypoints first. For instance, in an urban scenario with a 11 m/s (25 mph) speed limit, the ego-vehicle can utilize the first H/2 waypoints to formulate a trajectory that spans 15 m without needing to replan or generate a new estimate before the 1 s mark. Even though a motion planner would still execute collision checking as part of a downstream process, this illustrates the potential in real-time applications for urban driving. In fact, we find that our approach can keep up with real-time compute requirements by achieving an inference time of approximately 6.18 ms using the attention-based encoder and 6.22 ms using the CNN-based encoder. As an added benefit, the self-attention mechanism uses 31% fewer parameters than the CNN-based encoder, which comprises 16.8 M learnable parameters.

### 4.5. Discussion

A number of visualizations from various intersection scenarios are shown in Figure 6. These visualizations are generated using the SPF graph-based model and represent prediction outputs from three-way and four-way intersections that capture the multimodal properties of the CVAE. An important benefit of this approach is that the trajectories generated from the formulation can be quickly evaluated based on their semantic map projections and determine if the trajectories are adequate candidates for downstream motion planning. Nevertheless, the benchmarks indicate that potential improvements can be performed for longer horizon predictions as shown in Figure 7. The performance gap observed in this scenario can be the result of the complex intersection configuration that falls outside of the training dataset as the road geometry is unique. These performance gaps require further investigation for future work. Nevertheless, given that fully dynamic methods are inherently challenging due to the drastic variations in road topologies and occlusion, another research direction can entail unifying traditional planning stacks with fully dynamic path generation methods. This research direction can leverage map change detection similar to [40] to determine if a conventional stack should shift from using mapped futures to dynamic path generation methods similar to ours. This work additionally enables automatic updates to offline maps without hindering the performance of an autonomy stack in the presence of outdated mapped features.

The mainstream strategies rely on HD maps for planning [11,12] or explore methods to automate map generation [4,6,7,8] and maintain the HD map by change detection [40]. We argue that a more scalable substitution for HD maps is a combination of coarse maps and local environment models. Our work explores leveraging a coarse map and local semantic map for planning. We break down the planning system into global planning, local reference path generation, and dynamic motion planning and focus on the reference path generation aspect. While our approach is a step towards planning free of HD maps, further investigation is needed in various aspects. First, our work does not consider the dynamic motion planning aspect, which currently also relies on HD maps that provide the lane boundaries position/type, speed limit, and traffic control. Second, the local semantic map is built from fusing point clouds and images. In the case of using a sparse LiDAR to provide the point cloud (e.g., VLP-16), pre-map the environment to ensure point density is necessary [32]. Last but not least, an important aspect of planning is its interpretability, which relates to whether the method can be certified. Further investigation on the interpretation and certification [41] of machine learning based planning is needed for large-scale adoption.

## 5. Conclusions

In this work, a method for aligning coarse global plan representations with semantic scene models was explored. Various datasets and benchmarks with open implementations for the planner and CVAE formulation were made available. The contributions indicate potential use cases for urban driving navigation with fewer map priors, such as the widely used HD maps. This additionally presents directions for unifying dynamic path generation strategies with existing frameworks that leverage offline maps in combination with map detection methods. For future work, we plan to further reduce the complexity of the semantic scene model and evaluate the performance of such strategies in real-time autonomous driving software architectures.

## Figures and Tables

**Figure 1 sensors-23-06764-f001:**
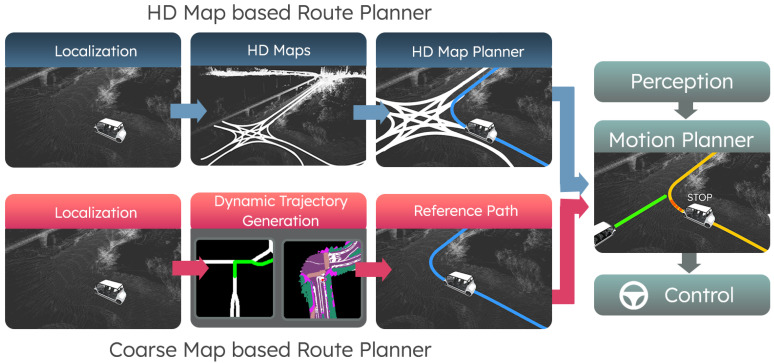
The visualization outlines the differences between traditional HD map based planners (blue modules) and our proposed approach (red modules). Both planning strategies can characterize reference trajectories that are amenable for downstream tasks (green modules).

**Figure 2 sensors-23-06764-f002:**
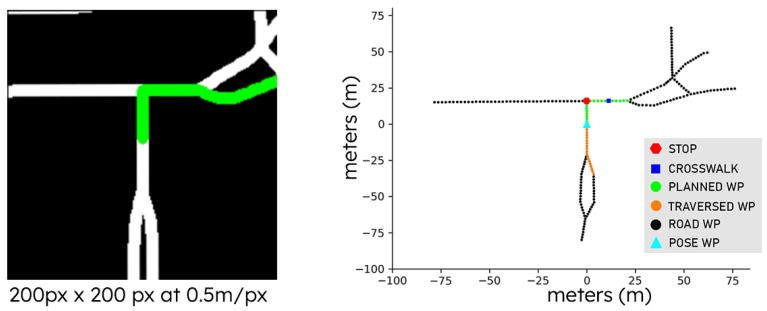
Global plan representations generated by our open-source navigation package. On the left, an image-based representation is shown and on the right, the graph-based representation is shown with various road features such as stop signs, crosswalks, and information about the history and planned trajectories.

**Figure 3 sensors-23-06764-f003:**
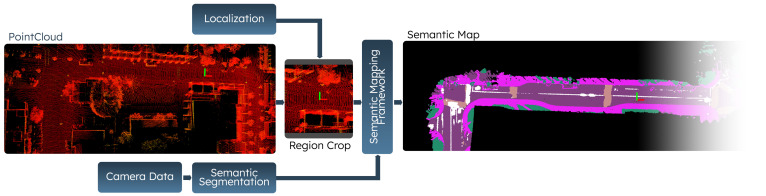
The figure represents the semantic mapping framework utilized to generate large 2D BEV semantic maps for the UC San Diego campus. It is composed of a camera-LiDAR fusion strategy that can aggregate multiple estimates across time and perform updates. The automatically generated maps can then be used with localization to provide egocentric representations for navigation applications.

**Figure 4 sensors-23-06764-f004:**
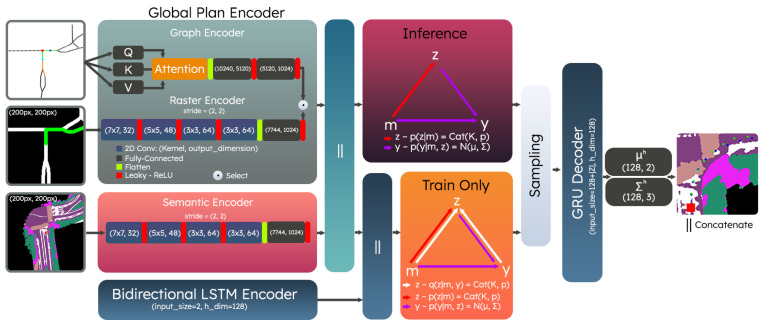
The CVAE network architecture is presented with the various components. The global plan encoders can process graph-based or rasterized representations. Similarly, the semantic scene representation is encoded using a sequence of CNN layers. Both representations are jointly used as conditionals in the training and test process.

**Figure 5 sensors-23-06764-f005:**
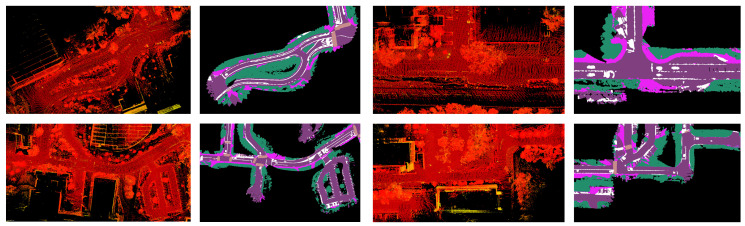
The visualizations in a map frame perspective correspond to the pointcloud maps used to generate 2D BEV semantic maps for various parts of a university campus. The pointcloud maps are shown in the first and third columns, and the associated 2D semantic maps are shown in the second and fourth columns. The camera-LiDAR fusion process outlined in Section 3.2.

**Figure 6 sensors-23-06764-f006:**
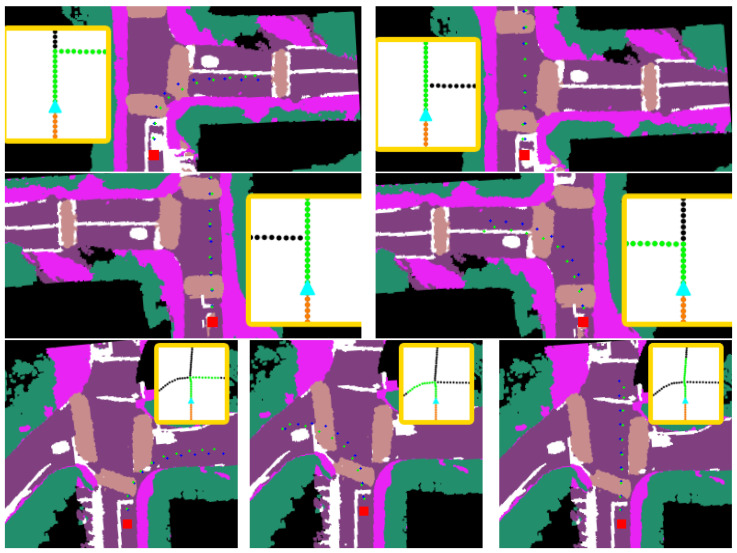
Various visualizations from the IntersectionScenes dataset using the SPF graph encoder. Each row represents an intersection with different global plans (yellow boxes). Groundtruth labels are denoted by blue waypoints and predictions by green waypoints. The red square corresponds to the rear-axle of the ego-vehicle.

**Figure 7 sensors-23-06764-f007:**
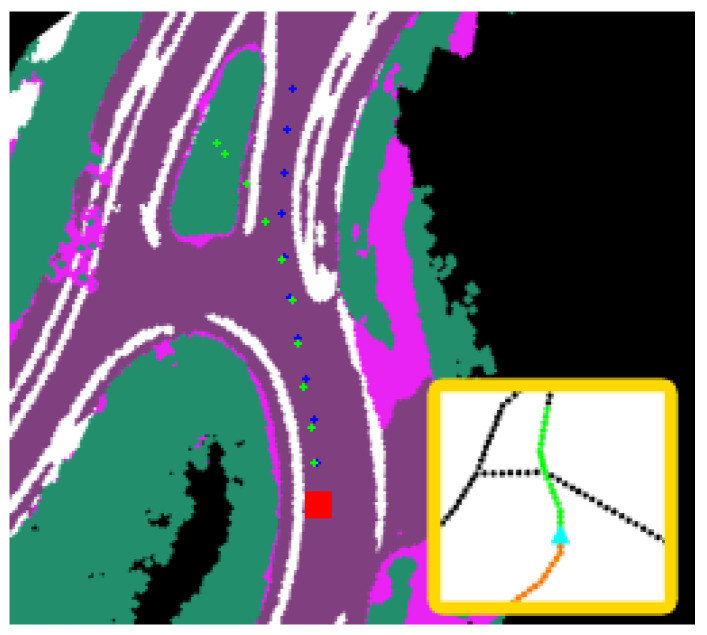
Longer horizon failure mode using the SPF graph encoder. While near horizon keypoint prediction appears adequate with low error, the predictions in the last 9 m deviate from groundtruth. The red square corresponds to the rear-axle of the ego-vehicle.

**Table 1 sensors-23-06764-t001:** The comparison between the H10 and H15 models is presented. Rasterized global plan representations are used to evaluate the performance of the models using the NominalScenes dataset. Error is given in meters, see Equations (Equation 5)–(Equation 11) for definitions. Bolded results indicate improvement.

Method	ADE${}_{\mathrm{FULL}}$ ↓	ADE${}_{\mathrm{HALF}}$ ↓	FDE ↓	MDE ↓
OSM-H10	1.056245	0.336941	2.447714	2.494614
OSM-H15	2.341875	0.753802	6.127183	6.177646

**Table 2 sensors-23-06764-t002:** The results of an ablation study between rasterized and graph-based representations are presented. These results include a comparison of the graph-based models as a function of different types of node features. The experiments are performed with the NominalScenes dataset. Error is given in meters, see Equations (Equation 5)–(Equation 11) for definitions and details on the indicator function to estimate drivable area compliance (DAC). Bolded results indicate improvement.

Method	ADE${}_{\mathrm{FULL}}$ ↓	ADE${}_{\mathrm{HALF}}$ ↓	FDE ↓	MDE ↓	DAC ↑	DAC${}_{\mathrm{HALF}}$ ↑
OSM Raster	1.056245	0.336941	2.447714	2.494614	0.849162	0.934218
OSM ATT w/PF	1.365685	0.538815	2.852894	3.047669	0.892321	0.933054
OSM ATT w/SPF	0.969206	0.353576	2.316740	2.393168	0.914869	0.944642
OSM ATT w/SCPF	1.131581	0.388303	2.717636	2.795832	0.905864	0.942408

**Table 3 sensors-23-06764-t003:** The graph-based and rasterized plan representations are evaluated an intersection-specific dataset, IntersectionScenes. This includes three-way and four-way intersections for urban driving scenarios specifically. Error is given in meters, see Equations (Equation 5)–(Equation 11) for definitions and details on the indicator function to estimate drivable area compliance (DAC). Bolded results indicate improvement.

Method	ADE${}_{\mathrm{FULL}}$ ↓	ADE${}_{\mathrm{HALF}}$ ↓	FDE ↓	MDE ↓	DAC ↑	DAC${}_{\mathrm{HALF}}$ ↑
OSM Raster	1.793062	0.673450	3.672231	3.728120	0.858367	0.913147
OSM ATT w/SPF	1.511415	0.619056	3.087722	3.226302	0.898473	0.924834

## Data Availability

The code, benchmark, and dataset documentation can be found under https://github.com/AutonomousVehicleLaboratory/coarse_av_cgm.git, accessed on 26 July 2023. We additionally make our OSM planner available: https://github.com/AutonomousVehicleLaboratory/gps_navigation, accessed on 26 July 2023.

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
