# Peer review of "Conditional Generative Models for Dynamic Trajectory Generation and Urban Driving"

_sensors, 2023, doi:10.3390/s23156764_

Round 1

Reviewer 1 Report

In this work, the authors presented a method for dynamic trajectory generation in urban driving environments when HD maps are not available but only global planning on OSM is available.

To incorporate high-level instructions (i.e. turn right vs. turn left at intersections), they formulate a conditional generative model strategy to explicitly capture the multimodal characteristics of urban driving. To evaluate the performance of the proposed method, a dataset is collected, labeled and made publicly available using multiple full-scale vehicles. 

A few suggestions: 

1. Data tables can be improved: add units when available; add more descriptions in table captions, for example Table 1 and Table 2.

2. A few figures are using full page width (Fig. 2, 3, 4 A1) but some others are using text  width (Fig. 1, 5). Check with format guidelines for consistent formatting.

Method section can be improved by reduce the portion of background information (maybe move it to related work) and expand the CVAE with more details

Author Response

Thank you for providing feedback on our paper; we sincerely appreciate your input. Please see our point by point responses below.

Point1: Data tables can be improved: add units when available; add more descriptions in table captions, for example Table 1 and Table 2.

Response2: We have incorporated more descriptions to the figures, specifically Figure A1. Tables 1, 2, 3 now explicitly include units. The reader is also referred to Appendix B.1 for more details on the equations and metrics.

Point2: A few figures are using full page width (Fig. 2, 3, 4 A1) but some others are using text  width (Fig. 1, 5). Check with format guidelines for consistent formatting.

 Response2: We are using the template provided by MDPI. Some of the tables in the template are half size and other tables are full size. The use of half vs full size tables was determined based on the content; for example, Tables 2 and 3 incorporate additional metrics and as a result require more space. For this reason, we utilized full size only when necessary. Moving forward, we are open to suggestions from the editors to ensure it fits the standard expected.

Reviewer 2 Report

In this work, the methodologies for dynamic trajectory generation for urban driving environments by utilizing coarse global plan representations are investigated. To incorporate high-level instructions (i.e. turn right vs. turn left at intersections), the authors compared various representations provided by lightweight and open-source OpenStreetMaps (OSM) and formulated a conditional generative model strategy to explicitly capture the multimodal characteristics of urban driving. To evaluate the performance of the models introduced, a data collection phase is performed using multiple full-scale vehicles with ground truth labels. The methodology is novel and the validation is solid to some extent. I suggest an acceptance after some modifications. My comments are listed below:

- Please highlight the novelties or contributions of this work in the introduction in a comparable manner.

- Please provide a subsection that shows the details of the requirements of the hardware to implement the methodology proposed in this paper.

- The work in this paper is interesting. Existing related works have been presented to refine the trajectories based on the sensors on the automated vehicles such as: an automated driving systems data acquisition and analytics platform; autonomous vehicle kinematics and dynamics synthesis for sideslip angle estimation based on consensus kalman filter; automated vehicle sideslip angle estimation considering signal measurement characteristic. Please include them in the paper and discuss the pros or cons of them. For instance, the work an automated driving systems data acquisition and analytics platform is the first real-world research that systematically processes the sensor data in connected automated vehicles to localize and extract the trajectories of vehicles with high accuracy.

- When providing the results, how the ground truth data is obtained? Is that labeled by human or automatic software?

-The resolution of figure 1 can be improved. 

Author Response

Thank you for providing feedback on our paper; we sincerely appreciate your input. Please see our point by point responses below.

Point 1: Please highlight the novelties or contributions of this work in the introduction in a comparable manner. 

Response1: We have performed a number of changes to the introduction. We summarized the challenges and the limitations of the prior work and listed contributions of our work in the form of bullet points.

Point2: Please provide a subsection that shows the details of the requirements of the hardware to implement the methodology proposed in this paper.

Response2: We have added a new section (Section 4.2) to provide additional details on hardware requirements.

Point3: The work in this paper is interesting. Existing related works have been presented to refine the trajectories based on the sensors on the automated vehicles such as: an automated driving systems data acquisition and analytics platform; autonomous vehicle kinematics and dynamics synthesis for sideslip angle estimation based on consensus kalman filter; automated vehicle sideslip angle estimation considering signal measurement characteristic. Please include them in the paper and discuss the pros or cons of them. For instance, the work an automated driving systems data acquisition and analytics platform is the first real-world research that systematically processes the sensor data in connected automated vehicles to localize and extract the trajectories of vehicles with high accuracy.

Response3: Autonomous vehicles in urban environments require contextual information for navigation. For example, intersection navigation requires information about the road layout and a sense of direction. Should we drive straight or go right? To provide a better explanation of our contribution, we make comparisons with respect to traditional software stacks that utilize HD maps for similar tasks. We additionally outline how our approach can integrate with the motion planning and control components in Figure 1, which can help better understand where the vehicle specific dynamics and kinematics would fall into place. While vehicle specific dynamics and kinematics models are out of the scope of our contribution, we have incorporated the recommended data acquisition work as a reference in the groundtruth data generation process.

Point4: When providing the results, how the ground truth data is obtained? Is that labeled by human or automatic software?

Response4We added description to the groundtruth generation process to section 4 Experiments and Data: “We use the reported ego-vehicle poses from a LiDAR based registration method to characterize groundtruth trajectories.” We have provided a figure on the appendix (Figure A1) where we describe our platforms and outline the data collection process. This figure is referenced in the text.

Point5: The resolution of figure 1 can be improved.

Response5: The resolution of figure one (now figure 2) appears to be appropriate. However, the resolution of the OSM representation on the left is indeed “low resolution” which is used to encode a global plan with fewer priors. In contrast to HD maps, our research contribution focuses on using coarse maps instead of high definition representations to minimize the cost associated with dense representations that can quickly become outdated in highly dynamic environments.

Reviewer 3 Report

Conditional Generative Models for Dynamic Trajectory

Generation and Urban Driving

This work explores methodologies for dynamic trajectory generation for urban driving 

environments by utilizing coarse global plan representations. In contrast to state-of-the-art architec- 

tures for autonomous driving that often leverage lane-level high-definition (HD) maps, we focus 

on minimizing required map priors that are needed to navigate in dynamic environments that may 

change over time.

- In academic work, comparing the obtained results to some related/recently published works under the same conditions (i.e., databases + protocols of evaluation) is necessary. The objective is to show the superiority of the presented work against the existing ones. Please explain more about the previous research result in this field.

- This paper major contributions is not clear, it is suggest to rebuttal rewrite the manuscript again. 

-Please follow the journal template in your next submission

- Need to add future study in the conclusions. 

- This paper does not answer the primary purposes of this paper that explain in the introductions.

- Need to discuss the proposed method's benefits and limitations.

Author Response

Thank you for providing feedback on our paper; we sincerely appreciate your input. Please see our point by point responses below.

Point1: In academic work, comparing the obtained results to some related/recently published works under the same conditions (i.e., databases + protocols of evaluation) is necessary. The objective is to show the superiority of the presented work against the existing ones. Please explain more about the previous research result in this field.

Response1: In the introduction we have provided additional information on how our work compares to existing methods and further explain the importance of exploring lighter representations, which provide benefits in terms of scalability and maintenance costs. We have added Figure 1 to help emphasize the differences between our approach and existing HD map methods. At the same time, it should help illustrate the benefits of having an approach that is not platform specific and can be integrated with downstream components such as perception, motion planning and control. Some of our key contributions include two datasets and our code repositories for the experiments. The tasks introduced in this work are underexplored and as a result, finding datasets and related works that can be evaluated fairly is non trivial. For this reason, we found that making our data and code public was necessary.

Point2: This paper major contributions is not clear, it is suggest to rebuttal rewrite the manuscript again.

Response2: We have performed a number of changes. We summarized the challenges and the limitations of the prior work and added clear contributions of our work in the form of bullet points.

Point3: Please follow the journal template in your next submission

Response3: Our original and revised submissions utilize the template from MDPI; this template includes figure and table templates. The final submission is likely to be reformatted by MDPI editors.

Point4: Need to add future study in the conclusions. 

Response4: Our conclusion includes directions for future work. We additionally highlight a number of takeaways that could help in future directions in the discussion section.

Point5: This paper does not answer the primary purposes of this paper that explain in the introductions.

Response5: We have rewritten the introduction section to not only highlight the contributions of our work but additionally compare to existing HD map based strategies. Figure 1 now illustrates these differences.

Point6: Need to discuss the proposed method's benefits and limitations.

Response6: We have added more details on the discussion section to explicitly cover benefits and limitations. The benefits of lightweight scene representations are additionally emphasized in the introduction and motivation for this work.

Round 2

Reviewer 2 Report

I have no further questions. The paper can be accepted. 

Reviewer 3 Report

The author revised the manuscript based on my comment this paper can be accepted.